# Sucrosomial Iron Supplementation for the Treatment of Iron Deficiency Anemia in Inflammatory Bowel Disease Patients Refractory to Oral Iron Treatment

**DOI:** 10.3390/nu13061770

**Published:** 2021-05-22

**Authors:** Guillermo Bastida, Claudia Herrera-de Guise, Alicia Algaba, Yolanda Ber Nieto, Jose Manuel Soares, Virginia Robles, Fernando Bermejo, Esteban Sáez-González, Fernando Gomollón, Pilar Nos

**Affiliations:** 1Department of Gastroenterology, CIBEREHD, Hospital Universitari i Politècnic La Fe, 46026 Valencia, Spain; esteban.digestivo@gmail.com (E.S.-G.); pilarnos@gmail.com (P.N.); 2Crohn-Colitis Care Unit, Vall d’Hebron Hospital Universitari, 08035 Barcelona, Spain; cherreradeguise@gmail.com (C.H.-d.G.); virgiroblesalonso@gmail.com (V.R.); 3Department of Gastroenterology, Instituto de Investigación Sanitaria Hospital La Paz (IdiPaz), Hospital Universitario de Fuenlabrada, 28046 Madrid, Spain; alicia_algaba@hotmail.com (A.A.); fernando.bermejo@salud.madrid.org (F.B.); 4Department of Gastroenterology, Hospital Universitario San Jorge, 22004 Huesca, Spain; ybernieto@gmail.com; 5Department of Gastroenterology, Hospital Pedro Hispano, 4454-509 Matosinhos, Portugal; jmmgso@gmail.com; 6IBD Unit, Digestive Diseases Service, Instituto de Investigación Sanitaria de Aragón (IIS), CIBEREHD, Hospital Clínico Universitario “Lozano Blesa”, 50009 Zaragoza, Spain; fgomollon@gmail.com

**Keywords:** inflammatory bowel disease, anemia, iron deficiency, iron supplementation, sucrosomial iron

## Abstract

Iron deficiency anemia (IDA) is a common manifestation of Inflammatory Bowel Disease (IBD). Oral iron supplements are the treatment of choice, but are not always well tolerated. Sucrosomial^®^ iron (SI) may represent an alternative. This prospective study assessed the tolerability and effectiveness of SI, and quality of life (QoL) of IDA-IBD patients who were intolerant to oral iron salts. The study included 52 individuals treated with 1 capsule/day for 12 weeks. Tolerability was assessed through a gastrointestinal symptom severity questionnaire. Hemoglobin (Hb) levels and clinical symptoms of IDA were analyzed. QoL was assessed using IBDQ-9 and EuroQoL questionnaires. The percentage of patients with excellent/good health increased from 42.9% to 94.3%. Mean Hb concentration significantly increased at all follow-up visits (*p* < 0.05). Almost all participants (96.9%) were adherent to the study medication. Patients’ QoL improved (IBDQ-9: from 60.9 to 65.5). Patients also improved in mobility (71.8% to 78.1%), usual activities (51.3% to 68.7%), pain/discomfort (41.0% to 53.1%), and extreme depression/anxiety problems (7.7% to 3.2%); they worsened in self-care (100% to 90.6%), but perceived an enhancement in their global health [EQ-VAS score: 61.9 (±26.1) to 66.9 (±20.3)]. SI was well tolerated and improved IDA symptoms, IBD activity, and patients’ QoL. In conclusion, SI should be considered in IDA–IBD patients.

## 1. Introduction

Inflammatory bowel disease (IBD) is an inflammatory condition of multifactorial etiology that describes a set of chronic gastrointestinal illnesses, including Crohn’s disease (CD) and ulcerative colitis (UC). One of the most common systemic complications in patients with IBD is anemia, occurring more frequently among women and in patients with CD [1,2,3]. The prevalence of anemia in IBD is still uncertain and ranges from approximately 20% to 70%, depending on both the definition of anemia and the study populations —e.g., hospitalized patients or outpatients [4]. IBD-associated anemia can be multifactorial in origin, but is typically caused by a combination of iron deficiency anemia (IDA), anemia of chronic disease (ACD) or inflammatory anemia, and mixed anemia (both IDA and ACD) [5,6]. Anemia in these patients results from functional or absolute iron deficiency, as a consequence of poor dietary intake of iron, decreased absorption (in part as a result of inflammation) and/or blood loss [2,7]. Other less frequent causes include vitamin B12 and folate deficiency, and IBD-related medication (such as thiopurines and sulfasalazine) [2,8]. Rare causes of IBD-anemia are hemolysis, myelodysplastic syndrome, and chronic renal insufficiency, among others [9]. Despite its considerable prevalence in patients with IBD, iron deficiency anemia (IDA) frequently remains underdiagnosed and undertreated because of the vague and nonspecific overall symptoms, that can be difficult to distinguish from the symptoms of the underlying disease [10,11]. Besides, IDA has a significant impact on physical condition, cognitive function and quality of life (QoL) and requires a prompt therapeutic intervention [8,12,13]. 

The available treatment options include oral and intravenous iron therapy, aimed at normalizing hemoglobin levels and replete iron stores [14,15]. Oral iron supplements, provided as ferrous or ferric salts, are usually the standard first-line treatment in patients with IDA because of their availability, ease of administration, and relatively low cost [14]. Nonetheless, they are limited in certain gastrointestinal conditions, due to their low bioavailability and poor tolerability. Particularly, oral iron therapy in IBD may be inappropriate in cases of severe IDA, in which rapid iron repletion is required, but is recommended in patients with mild anemia, whose disease is clinically inactive and who have not been previously intolerant of oral iron [11,15].

To increase tolerability and enhance intestinal absorption, and consequently reduce dosage and side effects, new oral iron supplements have been developed [16,17]. Among them, Sucrosomial^®^ iron (SI) represents an innovative oral formulation in which ferric pyrophosphate is protected by a phospholipid bilayer membrane, composed mainly from sunflower lecithin, plus a sucrester matrix (sucrosome) [18,19,20,21]. Figure 1 schematically represents the mechanisms of absorption of both conventional iron and SI. The presence of the sucrester matrix confers gastro-resistant properties to SI, allowing the intact sucrosome to reach the intestinal mucosa. In the intestine, SI can be absorbed across intestinal epithelium as a vesicle-like structure through M cells, paracellular and transcellular routes, bypassing the conventional iron absorption pathway and without the need for specific transporters [22]. Subsequently, SI is incorporated into the lymphatic system [19,22]. This mechanism confers an increased bioavailability of iron from SI, reduces gastrointestinal toxicity, thanks to the absence of direct contact with the intestinal mucosa, and prevents iron instability in the gastrointestinal tract. As a result, this method of iron supplementation has been associated with a high gastrointestinal absorption and bioavailability with a low incidence of side effects, being more efficacious and tolerable than other oral iron salts [18,19]. Thus, oral SI emerges as an alternative option for treating IDA in IBD patients who are unresponsive to, or intolerant to, iron salts. 

However, analysis of effectiveness and adverse effects of SI have not yet been extensively investigated in IBD and deserve further examination. Moreover, studies exploring the impact of this supplement on the health-related QoL of patients are lacking. Therefore, we designed a prospective study to assess the tolerability and effectiveness of SI in patients with IBD, as well as to analyze its influence on patients’ QoL. 

## 2. Materials and Methods

### 2.1. Study Design and Patient Population 

This was a prospective, multicenter, observational study conducted in four centers in Spain and one in Portugal, primarily aimed at evaluating the tolerability and effectiveness of SI in IBD, and the QoL of patients. Inclusion criteria were patients aged ≥18 years with a confirmed diagnosis of UC or CD; patients with hemoglobin (Hb) levels 10 < Hb < 13 g/dL in males, and 10 < Hb < 12 g/dL in females; serum ferritin <30 μg/L (iron deficiency), and transferrin saturation (TSAT) <16%; patients were also required to have previously failed on treatment with, or to be intolerant to, other oral iron formulations, and thus were appropriate candidates to receive SI supplementation. IBD patients with other hematologic disorders, blood transfusions within 30 days upon enrollment, renal insufficiency or with poorly controlled comorbidities (diabetes, heart failure, COPD) were excluded from the study. All individual participants received written and verbal information concerning the study and provided signed written informed consent. The study was approved by the local Ethics Committees and conducted in accordance with the Declaration of Helsinki. 

### 2.2. Treatment Plan and Assessments

The treatment consisted of one oral capsule per day of SI—containing 30 mg of elemental iron carried through the sucrosome—for 12 weeks. Each patient completed three-doctor visits: at baseline (V1), at week 5 (V2), and at week 13 (V3). Blood samples were collected at baseline and at weeks 4 and 12 after treatment initiation.

On the first visit, individuals who met the eligibility criteria were informed about the general principles of the study. All patients gave written informed consent. The demographic characteristics, medical history, and previous and current drug history of each patient were recorded. A routine clinical examination was also carried out for all participants upon admission. This examination included a complete physical examination and an evaluation of clinical symptoms and signs of IDA (such as asthenia, dyspnea, paleness, palpitations, tachycardia, tinnitus, poor concentration and irritability), as well as blood tests and relevant laboratory analyses measuring Hb, hematocrit, mean corpuscular volume (MCV), mean corpuscular hemoglobin (MCH), erythrocytes, ferritin, TSAT, sideremia, transferrin, and C-reactive protein. Patients were also questioned to determine disease activity in UC with the modified Mayo Score (MS) scale (remission = 0; mild disease = 1; moderate disease = 2; severe disease = 3) [23,24] and in CD with the Harvey–Bradshaw Index (HBI) (HBI ≤ 4 remission, 5–7 mild activity, 8–16 moderate activity, >16 severe activity) [25]. The impact on QoL was measured using two questionnaires: the shortened and validated version of the IBD Questionnaire, which only contains one domain (total score) (IBDQ-9) [26], and the EuroQoL [27]. The EuroQoL comprises a descriptive health classifying system (EQ-5D) with five dimensions (mobility, self-care, usual activities, pain/discomfort and anxiety/depression; each dimension had three levels: no problems, moderate problems and extreme problems) and a visual analogue scale (EQ-VAS) that allows respondents to provide a global assessment of their health status “today”, from 0 to 100. After all evaluations, subjects were instructed to take SI, once a day, for 12 weeks.

Patients received telephone-based monitoring after one week, and the investigator asked the patient about the tolerability of the treatment through a questionnaire (see “Outcomes” for more detail).

Four weeks after the first visit, blood tests and laboratory analyses were performed. At V2, the results of the laboratory analyses were evaluated; other evaluations included a complete physical examination; clinical symptoms and signs of iron deficiency; tolerability of the study medication; incidence of adverse events and patient adherence.

At week 12, blood tests and laboratory analyses were performed. The last study visit occurred at week 13, in which gastroenterologists performed the same evaluations carried out during the second visit. In addition, patients were questioned to determine their QoL with IBDQ-9 and EuroQoL questionnaires. The extent of patient compliance was also evaluated and calculated as the ratio of the number of doses taken to the number of doses prescribed. Satisfactory adherence was defined as >80% compliance with the prescribed dose regimen.

### 2.3. Outcomes

The primary outcome was to assess the tolerability of oral SI through a questionnaire that included questions regarding the patients’ general well-being (assessed as a single item rated on a 5-point scale: very good = 0; good = 1; fair = 2; bad = 3; very bad = 4), and the presence and severity of gastrointestinal symptoms normally associated with oral iron supplementation, such as diarrhea (assessed as number of liquid depositions per day, at least ≥1), abdominal pain, constipation, loss of appetite, nausea, vomiting, teeth staining, change in color of stool, and metallic taste (all these symptoms graded as none = 0; mild = 1; moderate = 2; severe = 3).

Secondary outcomes included an evaluation of the response to therapy by observing change in Hb (increase ≥ 2 g/dL) and/or normalization of Hb levels (>12 g/dL in women, >13 g/dL in males) by week 12. An evaluation of clinical symptoms and signs of IDA (asthenia, dyspnea, paleness, palpitations, tachycardia, tinnitus, poor concentration and irritability) between clinical visits was also performed. QoL was assessed using the shortened IBDQ-9 and EuroQoL questionnaires.

### 2.4. Statistical Analysis

All safety assessments were performed in the safety population, which included all randomized patients who received at least one dose of study test product. All effectiveness analyses were performed on the intent-to-treat (ITT) population. The ITT population consisted of patients who completed one-month treatment and who had blood tests and laboratory analyses performed at week 4.

Demographics and disease-related features were summarized descriptively using mean, standard deviation (SD), minimum and maximum, and absolute and relative frequencies, as appropriate. For all statistical analyses, a *p*-value < 0.05 was considered significant. The *p*-value for change from baseline was calculated with McNemar’s test, Student’s *t*-test or Wilcoxon signed-rank test, as appropriate. Cohen’s d was used to determine effect sizes. The statistical analyses were performed using SAS software, version 9.3 (SAS Institute Inc., Cary, NC, USA). 

## 3. Results

### 3.1. Baseline Characteristics and Participant Flow

A total of 52 patients were included in the study. Figure 2 shows the flow diagram for the study. Three patients were incorrectly selected and three more were excluded because they did not meet the inclusion criteria relative to Hb levels; these latest patients were under treatment for one week and thus were included in the safety population. Most patients (35/46, 76.1%) finished the 12-week treatment, and 23.9% (11/46) prematurely abandoned the study, due to adverse events (5); protocol violation (1); need for concomitant medication not permitted during the study (1); lost to follow-up (1); and voluntary withdrawal (3).

Baseline characteristics of the intention-to-treat population are shown in Table 1. The mean age of patients was 42 ± 12.0 years (range 20–81), with a mean body mass index (BMI) of 24.2 ± 5.1 Kg/m^2^ (10.9% of patients were obese with a BMI ≥30 Kg/m^2^); 67.4% of patients were diagnosed as CD and the rest were UC. In total, 34 patients (34/46, 73.9%) experienced symptoms of IDA; the most frequently reported symptoms were asthenia, present in 97.1% of patients with symptomatology, followed by poor concentration (55.9%) and irritability (44.1%).

### 3.2. Outcomes

#### 3.2.1. Tolerability of Sucrosomial^®^ Iron

The tolerability assessment of oral iron supplementation with SI throughout the study is presented in Table 2. According to the questions regarding the patients’ general well-being, less than half of patients rated their health as “very good” or “good” (42.9%) at baseline, whereas this percentage markedly increased at all follow-up time points (89.6% at 1 week, 100.0% at week 5, and 94.3% at week 13). These observed differences with respect to baseline values were statistically significant (*p* < 0.0001, in all cases). The same positive trend was observed for all gastrointestinal symptoms analyzed. Significant differences were found between the mean number of liquid depositions that patients presented at week 5 and at week 13, as compared to baseline (*p* < 0.0001 and *p* = 0.0003, respectively). Regarding abdominal pain, 46.9% patients reported having none or mild pain at baseline, and 92.5% and 88.6% at week 5 (*p* < 0.0001) and week 13 (*p* < 0.0001), respectively. Although constipation severity also decreased at all visits as compared to baseline, differences were not significant (*p* = 0.489 and *p* = 0.906, at V2 and V3, respectively). The number of patients suffering from loss of appetite decreased by 18.4% at V2 (*p* = 0.0082) and 9.5% at the end of the study (*p* = 0.1025). At baseline, 41.1% of patients reported nausea, 12.2% (*p* = 0.0076) at V2, and 11.4% (*p* = 0.0075) at V3. Changes observed in vomiting and teeth staining, despite improving across the study, were not statistically significant (*p* > 0.05). The color of stool statistically changed from baseline at all time visits (*p* = 0.0073 at V2; *p* = 0.0045 at V3). The sense of metallic taste was described in a low percentage of patients at V2 (12.2%) and V3 (11.4%), with respect to baseline (43.8%) (*p* = 0.0027 and *p* = 0.0016, respectively).

#### 3.2.2. Adverse Events (AEs)

Of all patients, 95.9% (47/49) experienced at least one AE during the study. The mean number of AEs per patient was 3.6 (range: 1–11), and the majority of participants (66.0%, 31/47) experienced four or less AEs. The events were mostly reported as mild (68.5%) and moderate (24.8%), and only 6.7% were considered severe. Table 3 shows the most frequent AEs reported and the relatedness of all AEs to SI treatment. The most frequent secondary effects described were the change in the color of stool (55.1% of patients), abdominal pain (42.9%), and diarrhea (40.8%). Almost half of the AEs reported were either not related (25.9%) or unlikely to be related (22.9%) to the study treatment, whereas 40.9% of AEs were considered certainly (7.2%) or probably related (33.7%) to the treatment. In 91.0% of the occurrences, there was no change in medication, and SI was only temporarily interrupted in 2.4% of AEs, and permanently interrupted in 6.6% of events. Overall, study medication was discontinued prematurely due to AEs in a total of five patients, who experienced diarrhea, anemia and epigastralgia, epigastralgia, subocclusive symptoms, and constipation (see Figure 2).

#### 3.2.3. Adherence to the Study Treatment

Adherence to SI treatment was good, since 96.9% of participants were adherent to the study medication.

#### 3.2.4. Analytical Parameters

Table 4 shows the mean concentrations of hematologic and iron deficiency biomarkers at baseline, V2 and V3. Absolute mean Hb concentration significantly increased at follow-up visits, as compared to baseline: from 11.2 (0.7) g/dL to 11.4 (0.8) g/dL at V2 (*p* = 0.0194); and 11.7 (1.1) g/dL (*p* = 0.0055) at the end of treatment. Significant differences between mean Hb concentration at weeks 4 and 12 were also observed (*p* = 0.0245). After 4 and 12 weeks, hematocrit levels also increased above the baseline: from 35.6 (2.0) to 36.5 (2.1) at V2, and 37.3 (3.0) at V3 (*p* = 0.0009 and *p* = 0.0016, respectively). MCV, MCH, erythrocytes, ferritin, TSAT and sideremia showed the same tendency, rising at the end of the study visit above baseline and V2, while transferrin levels and C-reactive protein concentration decreased throughout the study. When considering the response to oral iron therapy with SI, 25.7% of patients (9/35) had an increase in Hb ≥2 g/dL and/or normalization of Hb levels (>12g/dL in women, >13g/dL in males) by week 12.

#### 3.2.5. Signs and Symptoms of IDA

At baseline, 72.5% of patients (29/40) reported symptoms of iron deficiency, and this percentage decreased to 54.3% (19/35) after the 12-week treatment. The mean number of clinical symptoms of IDA at baseline was 2.4 (±2.0) and 1.2 (±1.5) at the end of the study (*p* = 0.0120). More than half of patients (52.5%) presented 3 or more clinical symptoms of IDA at baseline, whereas this percentage significantly decreased to 10% and 17.1% at V2 and V3, respectively. Asthenia was the most frequent symptom observed, present in 97.1% of patients, followed by difficulty in concentrating (55.9%) and irritability (44.1%).

#### 3.2.6. Quality of Life (QoL)

The IBDQ-9 score increased from 60.9 at baseline to 65.5 at week 13, indicating an improvement in patients’ QoL. The results obtained with the EuroQoL questionnaire are shown in Figure 3. According to these data, there were very few participants that reported extreme problems in any of the five dimensions of the questionnaire. Patients experienced an improvement in three dimensions at the end of the study, as compared to baseline: mobility (71.8% vs. 78.1%), usual activities (51.3% vs. 68.7%), and pain/discomfort (41.0% vs. 53.1%). The percentage of patients with severe depression and anxiety problems also decreased compared to baseline (from 7.7% to 3.2%). However, patients with self-care problems increased at the end of study (from 0% to 9.3%). In particular, one patient had problems washing and dressing himself, and two more were unable to wash and dress themselves. Nonetheless, EQ-VAS score was higher at V3 than at baseline (66.9 (±20.3) vs. 61.9 (±26.1), respectively), indicating that patients perceived a global improvement in their health status at the end of the study.

## 4. Discussion

In this prospective study, we investigated the effectiveness and tolerability of SI, an innovative oral iron-containing carrier, in the treatment of IDA in IBD patients who had previously failed to respond to, or had been intolerant to, other available oral iron formulations. We also assessed the effects of this therapy on the QoL of patients.

The unique structure of SI, which is different from other oral formulations, protects iron from the acid environment of the stomach, increases its absorption across the intestinal epithelium through alternative routes, and ensures high bioavailability and low gastrointestinal toxicity. In fact, most relevant evidence on the tolerability and efficacy of oral SI in different preclinical and clinical settings seems to support oral SI as a new valid option for iron supplementation, being more efficacious and tolerable than conventional oral iron salts [19]. It has been also considered as an alternative to IV iron for initial and/or maintenance treatment in different patient populations, such as celiac disease, cancer, bariatric surgery and chronic kidney disease [28,29,30,31,32]. A very recent prospective study comparing the effectiveness and tolerability of oral SI and intravenous ferric carboxy-maltose (FCM) in patients with UC in remission and mild-to-moderate anemia, has shown that SI and intravenous iron therapy have similar effectiveness and tolerability. The authors support the preferential choice of oral therapy with SI, because of its lower cost and high levels of adherence [33]. This is remarkable since, to date, the comparisons between conventional oral iron supplementation and intravenous iron have shown that intravenous iron is more effective and better tolerated, as reported in a systematic review and meta-analysis conducted by Bonovas et al. [34].

In our study, the tolerability assessment of oral SI through the use of questionnaires indicated a significant improvement in the general well-being of patients after treatment. At the end of the study, almost 95% of patients perceived their health status either as good or very good, as compared to less than half of patients at baseline. In addition, the presence and severity of the most frequent gastrointestinal symptoms, such as change in color of stool, abdominal pain, diarrhea, nausea, and metallic taste, was significantly reduced after SI treatment. Although 34% of the reported events were judged as being probably related, and 7% certainly related to the supplement, the vast majority of events were of mild to moderate intensity and only five patients discontinued study treatment prematurely due to an AE. It is also important to highlight that 96.9% of participants that completed the study were adherent to the study treatment. These data compare favorably with a previous study assessing SI formulation in patients with IBD [18]. Outcomes corroborate the good tolerability of the 12-week treatment with SI in IBD patients previously unresponsive or intolerant to traditional oral iron salts.

Regarding effectiveness, the low dose of SI tested in the study (30 mg/day) was significantly effective in improving Hb levels. In fact, after treatment, a mean 0.5 g/dL increase in Hb levels was observed (*p* = 0.0055). The 12-week treatment also increased mean hematocrit levels and other iron parameters. Again, these results are in accordance with previous observations [18]. When considering the response to SI supplementation, 25.7% of patients had an increase in Hb ≥ 2 g/dL and/or normalization of Hb levels by week 12. Patients also experienced a reduction in IBD activity and an amelioration of the systemic inflammatory response, as shown by a decrease in transferrin levels and CRP concentration. Our results, therefore, indicate that SI is effective in treating IDA in patients with IBD. These outcomes also suggest that the use of higher doses of SI might enhance effectiveness without compromising safety, although a balanced profile in terms of effectiveness and tolerability should be explored. So far, the use of 30–60 mg/day of SI for 2–3 months in several case series of IBD patients with mild-to-moderate IDA has been shown to be efficacious in rising Hb concentrations, as well as ferritin and TSAT levels, with very few gastrointestinal side effects [19]. Furthermore, the reported symptoms of iron deficiency considerably decreased after 4 and 12 weeks of treatment, although the percentage of patients that presented three or more clinical symptoms at V2 was lower than at V3 (10% vs. 17.1%, respectively). Patients not responding to SI could have contributed to these differences, since an ineffective iron medication would have led to an increased number of symptoms of anemia over time.

Overall, our findings suggest that SI supplementation appears to be a well-tolerated and effective treatment for IDA in IBD patients’ refractory to traditional oral iron. Further randomized controlled studies would be needed to evaluate whether SI could be considered as an alternative to intravenous iron in this population. Taking into consideration the conclusions of the previously mentioned study, that compares the effectiveness and tolerability of oral SI and intravenous FCM in UC patients [33], SI supplementation in both UC and CD patients could potentially constitute an equivalent treatment to intravenous iron therapy of IDA, in terms of effectiveness and tolerability.

The favorable treatment of iron deficiency observed in our study was accompanied by an improvement in the QoL of patients, as shown by the results obtained with both the IBDQ-9 and the EuroQoL questionnaires. The mean rating of the IBDQ-9 and the EQ-VAS score considerably increased at the last visit, indicating that patients perceived a global improvement in their health status after SI treatment. In all five dimensions of the EuroQoL questionnaire, there were very few participants that reported severe problems, and four of the dimensions (mobility, usual activities, pain/discomfort, and depression/anxiety problems) improved after treatment. The self-care domain, which was related to the ability of patients to wash and dress themselves, worsened at the end of the study in three patients. However, it is important to highlight that the EuroQoL questionnaire is not disease-specific, and the patient’s perception of general well-being includes several aspects of life [35]. Therefore, the capacity for self-care could have responded to any health incident experienced by patients and might not be directly related to anemia. In general, our results are in agreement with multiple studies over recent years that have shown a strong correlation of the overall QoL in IBD patients with the correction of anemia, independently from IBD disease activity [36,37,38].

The present study has some limitations, such as the lack of a control group and the relatively small sample size. Another limitation of the study is the fact that the study included a high percentage of women, and both CD and UC patients, with different grades of disease severity, thus affecting the generalization of results to all IBD patients. Subgroup analyses might help to address this, although they would be limited by the size of our population. Nonetheless, this study provides favorable outcomes supporting the use of SI supplementation in IBD patients, who were unresponsive or intolerant to conventional oral iron formulas. Further studies would be needed. It also serves as a basis for a broader exploration in larger populations, with the inclusion of a control group that would allow for the comparison of SI to other oral or intravenous iron preparations.

## 5. Conclusions

The results of this study confirmed that SI is well tolerated in patients with IBD with previous intolerance, or lack of response, to conventional oral iron salts. Furthermore, the low dose of SI tested was significantly effective in improving IDA symptoms and patients’ QoL, without adversely affecting IBD activity. According to these results, SI should be considered as an alternative option for the treatment of IDA in IBD patients.

## Figures and Tables

**Figure 1 nutrients-13-01770-f001:**
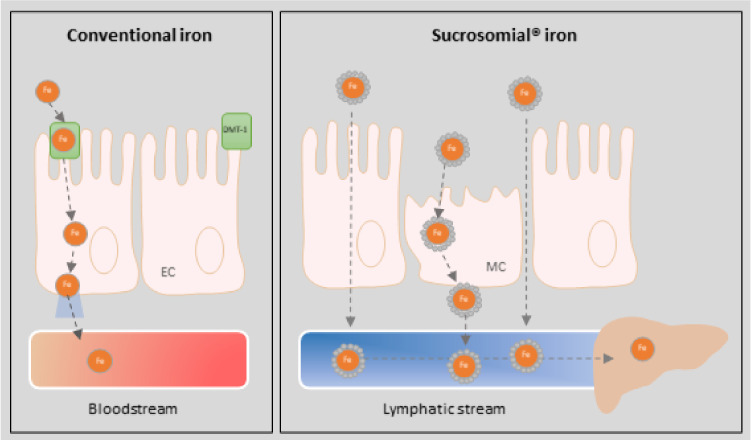
Schematic representation of absorption mechanisms of conventional iron and Sucrosomial^®^ iron (Pharmanutra Spa, Scientific Department, Pisa, Italy). EC, endothelial cells; MC, M cells.

**Figure 2 nutrients-13-01770-f002:**
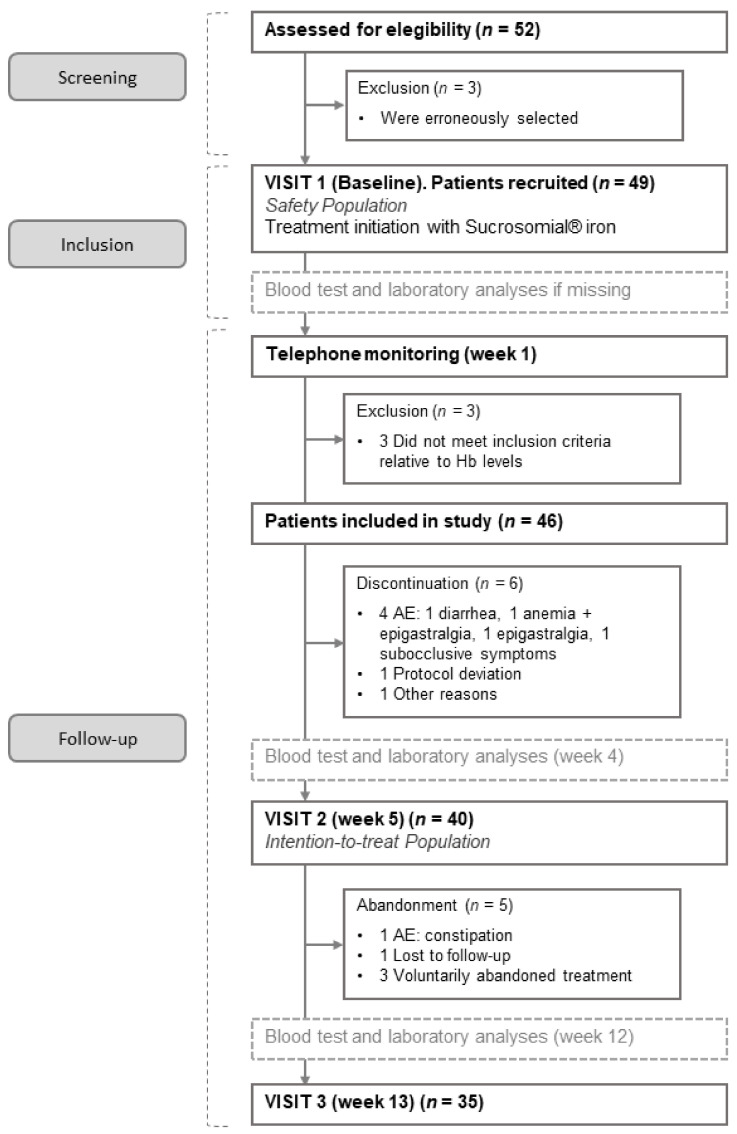
Flow diagram of patient selection for the study. AE: adverse event; Hb: hemoglobin.

**Figure 3 nutrients-13-01770-f003:**
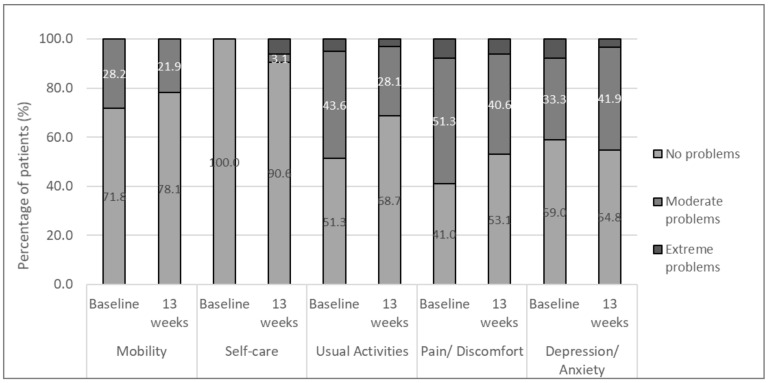
Distribution of responses on the EuroQoL dimensions.

**Table 1 nutrients-13-01770-t001:** Baseline characteristics of the intention-to-treat population (*n* = 36).

Characteristic	Value
Age, years [median (range)]	41.2 (20.0–81.2)
Sex, F/M [*n* (%)]	39 (84.8)/7 (15.2)
BMI, kg/m^2^ [median (range)]	23.5 (16.0–45.0)
<18.5	5 (10.9)
18.5–<25	26 (56.5)
2–<30	10 (21.7)
≥30	5 (10.9)
Diagnosis of CD/UC [*n* (%)]	31 (67.4)/15 (32.6)
Montreal disease extent in CD at diagnosis [*n* (%)]	
ileal (L1)	15 (32.6)
colonic (L2)	3 (6.5)
ileo-colonic (L3)	12 (26.1)
upper disease (L4)	0(0.0)
ileal + upper disease (L1 + L4)	1 (2.2)
Montreal disease extent in UC at diagnosis [*n* (%)]	
ulcerative proctitis (E1)	0 (0.0)
left-sided UC or distal UC (E2)	6 (13.0)
extensive UC or pancolitis (E3)	9 (19.6)
Modified Mayo Score (MS) [mean (SD)]	1.27 (1.75)
UC severity according to MS [*n* (%)]	
remission: 0	12 (80.0)
mild disease: 1	2 (13.3)
moderate disease: 2	1 (6.7)
severe disease: 3	0 (0.0)
Harvey-Bradshaw Index (HBI) score [mean (SD)]	2.8 (2.8)
CD severity according to HBI [*n* (%)]	
remission: ≤4	23 (76.7)
mild disease: 5–7	5 (16.7)
moderate disease: 8–16	2 (6.7)
severe disease: ≥16	0 (0.0)
Years since diagnosis of IBD [mean (SD)]	9.8 (7.0)
Patients with symptoms of IDA [*n* (%)]	34 (73.9)

BMI: body mass index; CD: Crohn disease; F: female; IBD: inflammatory bowel disease; IDA: iron deficiency anemia; M: male; UC: ulcerative colitis.

**Table 2 nutrients-13-01770-t002:** Tolerability of oral iron supplementation with Sucrosomial^®^ iron.

Parameters	Visit 1 (Baseline)	Telephonic Monitoring (Week 1)	Effect Size (Telephonic Monitoring)	Visit 2 (Week 5)	Effect Size (Visit 2)	Visit 3 (Week 13)	Effect Size (Visit 3)
General well-being [*n* (%)]	*n* = 49	*n* = 48	1.44	*n* = 41	0.93	*n* = 35	−0.18
very good health	8 (16.3)	31 (64.6)		28 (68.3)		17 (48.6)	
good health	13 (26.5)	12 (25.0)		13 (31.7)		16 (45.7)	
fair health	16 (32.6)	3 (6.2)		-		1 (2.9)	
bad health	10 (20.4)	2 (4.2)		-		1 (2.9)	
very bad health	2 (4.1)	-		-		-	
Number of liquid depositions [*n* (%)]	*n* = 48	*n* = 48	0.76	*n* = 41	1.44	*n* = 34	1.83
≤2	23 (46.9)	38 (79.2)		31 (75.6)		28 (80.0)	
3–5	16 (32.7)	5 (10.4)		9 (22.0)		5 (14.3)	
6–8	10 (20.4)	3 (6.2)		1 (2.4)		1 (2.9)	
≥9	-	2 (4.2)		-		1 (2.9)	
Abdominal pain [*n* (%)]	*n* = 49	*n* = 47	1.37	*n* = 40	1.71	*n* = 35	1.92
none	10 (20.4)	37 (78.7)		29 (72.5)		26 (74.3)	
mild	13 (26.5)	6 (12.8)		8 (20.0)		5 (14.3)	
moderate	20 (40.8)	3 (6.4)		2 (5.0)		3 (8.6)	
severe	6 (12.2)	1 (2.1)		1 (2.5)		1 (2.9)	
Constipation [*n* (%)]	*n* = 48	*n* = 47	0.84	*n* = 41	0.64	*n* = 34	0.74
none	37 (77.1)	44 (93.6)		35 (85.4)		29 (85.3)	
mild	5 (10.4)	2 (4.3)		5 (12.2)		4 (11.8)	
moderate	5 (10.4)	1 (2.1)		-		-	
severe	1 (2.1)	-		1 (2.5)		1 (2.9)	
Loss of appetite [*n* (%)]	*n* = 48	*n* = 46	1.06	*n* = 40	1.06	*n* = 35	0.79
none	38 (79.1)	46 (100.0)		39 (97.5)		31 (88.6)	
mild	6 (12.5)	- -		1 (2.5)		3 (8.6)	
moderate	3 (6.2)	-		-		1 (2.9)	
severe	1 (2.1)	-		-		-	
Nausea [*n* (%)]	*n* = 49	*n* = 48	1.05	*n* = 41	1.13	*n* = 35	1.25
none	29 (59.2)	44 (91.7)		36 (87.8)		31 (88.6)	
mild	12 (24.5)	3 (6.2)		4 (9.8)		3 (8.6)	
moderate	5 (10.2)	1 (2.1)		1 (2.4)		1 (2.9)	
severe	3 (6.1)	-		-		-	
Vomiting [*n* (%)]	*n* = 49	*n* = 48	0.99	*n* = 41	0.92	*n* = 35	0.81
none	44 (89.8)	48 (100.0)		40 (97.6)		33 (94.3)	
mild	3 (6.1)	-		1 (2.4)		2 (5.7)	
moderate	1 (2.0)	-		-		-	
severe	1 (2.0)	-		-		-	
Teeth staining [*n* (%)]	*n* = 48	*n* = 45	0.95	*n* = 41	0.44	*n* = 35	-0.06
none	45 (93.7)	45 (100.0)		39 (95.1)		32 (91.4)	
mild	2 (4.2)	-		2 (4.9)		2 (5.7)	
moderate	1 (2.1)	-		-		1 (2.9)	
severe	-	-		-		-	
Change in color of stool [*n* (%)]	*n* = 49	*n* = 47	1.40	*n* = 41	1.81	*n* = 34	2.09
none	11 (22.4)	35 (74.5)		22 (53.7)		20 (58.8)	
mild	3 (6.1)	7 (14.9)		14 (34.1)		8 (23.5)	
moderate	17 (34.7)	2 (4.2)		4 (9.8)		5 (14.7)	
severe	18 (36.7)	3 (6.4)		1 (2.4)		1 (2.9)	
Metallic taste [*n* (%)]	*n* = 48	*n* = 47	1.07	*n* = 41	1.09	*n* = 35	1.21
none	27 (56.2)	43 (91.5)		36 (87.8)		31 (88.6)	
mild	12 (25.0)	4 (8.5)		4 (9.8)		3 (8.6)	
moderate	9 (18.8)	-		1 (2.4)		1 (2.9)	
severe	-	-		-		-	

**Table 3 nutrients-13-01770-t003:** Description of most frequent adverse events reported and relatedness of all adverse events to Sucrosomial^®^ iron treatment.

**Description of Adverse Events**	**Number of Patients (%)** **(*n* = 49)**
Color of stool	27 (55.1)
Abdominal pain	21 (42.9)
Diarrhea	20 (40.8)
Nausea	10 (20.4)
Constipation	10 (20.4)
Metallic taste	8 (16.3)
Asthenia	7 (14.3)
**Relatedness to SI Treatment**	**Number of AEs (%)** **(*n* = 166)**
Certain	12 (7.2)
Probable	56 (33.7)
Possible	17 (10.2)
Unlikely	38 (22.9)
Not related	43 (25.9)

**Table 4 nutrients-13-01770-t004:** Hematological and biochemical parameters before and after oral iron supplementation with Sucrosomial^®^ iron.

Parameter [Mean (SD)]	Visit 1 (Baseline)	Visit 2 (Week 5)	Effect Size (Visit 2)	Visit 3 (Week 13)	Effect Size (Visit 3)
Hemoglobin (g/dL)	11.2 (0.7)	11.4 (0.8)	0.27	11.7 (1.1)	0.56
Hematocrit (%)	35.6 (2.0)	36.5 (2.1)	0.44	37.3 (3.0)	0.69
MCV (fL)	82.5 (7.2)	83.1 (7.8)	0.08	83.4 (7.9)	0.12
MCH (pg)	26.0 (3.0)	26.0 (2.9)	0.00	26.2 (3.2)	0.06
Erythrocytes (10^12^/L)	4.3 (0.7)	4.4 (0.4)	0.17	4.5 (0.5)	0.32
Ferritin (μg/L)	14.3 (14.0)	15.4 (12.8)	0.08	15.9 (13.0)	0.12
TSAT (%)	8.7 (4.4)	13.9 (11.2)	0.64	16.2 (15.9)	0.69
Sideremia (μg/dL)	31.4 (17.0)	47.9 (37.2)	0.59	55.6 (51.6)	0.68
Transferrin (g/L)	311.8 (66.4)	303.2 (64.0)	−0.13	298.7 (86.9)	−0.17
C-reactive protein (mg/L)	7.9 (9.0)	8.1 (13.2)	0.02	4.6 (7.3)	−0.40

MCV: mean corpuscular volume; MCH: mean corpuscular hemoglobin; TSAT: transferrin saturation.

## Data Availability

The data presented in this study are available in the article and archived in the Instituto de Investigación Sanitaria La Fe (Valencia, Spain).

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
