# Peer review of "Sucrosomial Iron Supplementation for the Treatment of Iron Deficiency Anemia in Inflammatory Bowel Disease Patients Refractory to Oral Iron Treatment"

_nutrients, 2021, doi:10.3390/nu13061770_

Round 1

Reviewer 1 Report

In the manuscript (Nutrients ID-1166622), the authors hypothesized that a novel oral iron formulation—Sucrosomial® iron (SI)—might be a good therapeutic option for patients with Inflammatory Bowel Disease (IBD) and iron deficiency anemia (IDA). The authors assessed the tolerability and efficacy of SI supplementation in 52 patients with IBD and IDA. Also, the authors assessed changes in Quality of Life (QoL) throughout this prospective study. The results support the authors’ hypothesis; study participants tolerated SI supplementation, which was substantiated with good compliance (96.9% of participants took at least 80% of prescribed SI regimen) and a decrease in gastrointestinal issues. Furthermore, the authors’ hypothesis was strengthened by the efficacy of SI supplementation. 25.7% of participants showed an adequate hematologic response (Hb increase >2 g/dL or Hb normalization) at week 13. Last but not least, the study showed an increase in general well-being during SI supplementation.

This manuscript is interesting because it brings attention to a significant and an overlooked issue in patients with IBD: (iron deficiency) anemia and iron therapy. The authors emphasize the need for novel oral iron formulations with better bioavailability and safety profiles for more effective and tolerable iron supplementation in IBD-population.

The manuscript has several strengths. Firstly, studies on anemia and iron supplementation in patients with IBD report varying results due to the published studies’ heterogeneous nature. Previous studies used different iron deficiency definitions or different biomarker cut-off points; studies also included outpatient and hospitalized patient populations. The authors of this manuscript used up-to-date and widely accepted iron deficiency definitions, which are in-line with the most current guidelines, i.e. the European Crohn’s and Colitis Organisation (ECCO) guidelines on anemia management.

Furthermore, the authors assessed IDA symptoms at the baseline and week 13; most existing evidence emphasizes biochemical results instead of focusing on patient-reported outcomes/symptoms. Also, the authors evaluated IDA symptoms and gastrointestinal complaints—e.g., abdominal pain, bloating, constipation, diarrhea—at the baseline and at week 13, which helps to distinguish pre-existing issues from side-effects of the SI supplementation.

Last but not least, QoL was evaluated throughout the study. Authors used validated questionnaires to emphasize relatively low QoL of patients with IBD and concomitant IDA, and improvement in QoL with oral iron supplementation.

The manuscript adds new information to the current knowledge; however, it has several weaknesses:
- The introduction section only briefly mentions the working mechanism of SI. It also just mentions better bioavailability. The manuscript would improve if a more elaborate explanation was provided. Also, a figure that depicts absorption of traditional oral formulations and SI would improve the clarity of the manuscript even more.

- Page 2–3, lines 97–98: the authors write that patients with “highly prevalent comorbidities” were excluded. The description of said exclusion criteria is too vague and difficult to understand: highly prevalent comorbidities in patients with IBD or the general population? Which comorbidities?

- Page 6, lines 200–211: sentences reporting gastrointestinal complaints/side-effects are hard to read and follow. For example, the authors write, “Nausea was absent in 59.2% patients at baseline and...”, “Metallic taste was absent in the vast majority of patients at V2 (87.8%) and V3 (88.6%), with respect to baseline (56.2%)...”. The clarity of the manuscript would improve if authors reported the presence of gastrointestinal symptoms instead of their absence. For example: 41.1% of patients reported nausea at baseline, 12.2% (p=0.0076) at visit 2, and 11.4% (p=0.0075) at visit 3.

- Results are presented with p-values only. The authors did not report effect sizes or (95%) confidence intervals, which would provide more information on the significance of the reported results. The STROBE and the CONSORT guidelines include effect sizes and confidence intervals as the standard of reporting.

- Table 4 includes information on the relatedness of (S)AE to SI therapy. Information on the type and frequency of (S)AEs is described in a text format, which is hard to read. For clarity, this information would be better presented in a table. Authors might choose to include two separate tables or to edit Table 4 and include information on (S)AEs and their prevalence in the study population.

- Page 9, lines 261–268: includes parallelism in the sentence structure, but the second sentence includes three negatives (however, without self-care and depression/anxiety problems, and decreased). “However, the number of patients without self-care and depression/anxiety problems decreased...”. This sentence is hard to read and understand. In my opinion, the authors meant to say that self-care and depression/anxiety related problems increased compared to the baseline.

- Figure 2 contains a mistake in labeling. Label says “baseline” and “13 months”. It should be 13 weeks instead of 13 months. 

- Page 10–11, lines 322–325 (conclusion): authors write, “Furthermore, the low dose of SI tested was significantly effective in improved IDA symptoms and IBD activity, as well as patient’s QoL.” The conclusion suggests that oral SI might improve IBD disease activity. This is confusing because oral iron is not prescribed for the treatment of IBD disease activity. If I understand correctly, the authors want to emphasize that SI did not exacerbate the IBD, contrary to the traditional oral iron formulations.

- General conclusions made by authors are more positive than the reported results. In the Discussion section, the authors report that Sucrosomial® iron could be an alternative to intravenous iron due to tolerability and efficacy of SI (25.7% of participants had an adequate hematologic response). The authors also mention the significant Hb increase. This is true; the results are statistically significant, but the reported mean Hb values show an increase of 0.2 g/dL from baseline to week 5 and 0.5 g/dL from baseline to week 13. Based on the reported mean ferritin values, patients were still iron deficient at week 13 (ferritin 15.9 µg/L at week 13; just 1.6 µg/L increase from baseline). Compare these results with 65.6% of patients with an adequate hematologic response to intravenous iron (reported in a systematic-review by Bonovas et al.). Besides, the authors report that 95.9% of participants (page 8) experienced at least one adverse event. Meanwhile, the reported adverse event rate is 2.4% in patients who received intravenous iron (Bonovas et al.). One might raise concerns if SI can be a viable alternative for intravenous iron rather than an alternative to traditional oral iron formulations.

Several interesting findings are reported in the Results section, but they are not addressed in the Discussion section:
- Study population includes a very large percentage (84.4%) of female participants. Are the results representative of all IBD patients?

- Page 9, lines 253–255: there was a relative increase in IDA symptoms at visit three compared to visit two (17.1% vs. 10%). The authors did not provide a possible explanation. Was the increase in IDA symptoms observed in participants with inadequate response to the SI supplementation? The authors do not provide a possible explanation.

- Page 9, lines 264–254: there was a slight increase in self-care and depression/anxiety problems at the end of the study. Can the authors explain these results? Was this increase in patients who did not respond to the SI supplementation? Or experienced the most (or the most severe) side-effects?

Additional suggestions for improvement:
- Page 3, lines 109–110: authors state that previous and current drug use was recorded. It would be interesting to see if the response to the SI supplementation might have been influenced by the use of IBD unrelated medication, such as proton-pump inhibitors (if such information is available).

- This study shows that the SI supplementation does not exacerbate IBD. If available, information on fecal calprotectin measurements at baseline and at the end of the follow-up period would substantiate the findings even more.

- Adequate compliance was defined as the use of at least 80% of the prescribed SI dosage (pills). The authors report that 96.9% of patients showed good compliance. It would be interesting to see if there were participants who showed even better compliance, e.g. took at least 90% of the prescribed doses or even 100%. 

Reviewer 2 Report

This is an interesting study evaluating sucrosomial iron  (SI) therapy in IBD. Although several studies evaluated this treatment in different chronic conditions, few data are available in IBD setting, and an analysis of Quality of Life was never been performed in this setting, to the best of my knowledge.

1) In Introduction section, the classification of the different types of anemia in IBD should be added (iron deficiency, anemia related to chronic diseases, mixed type etc)

2) In Methods section, the number of patients included in the study (52) should not be inserted, since it is a result. Methods section should only describe how patients were selected for the study. For this reason, it is unclear why three patients were incorrectly selected and three more were excluded because they did not meet the inclusion criteria relative to Hb levels... Since they have different characteristics at baseline, as well as they were treated for only one week with SI, I think that it is incorrect to include them in the safety analysis.

3) Disease extension should be showed in Table 1, with particular regard for patients with CD: an extensive ileal involvement should be considered as an exclusion criterion, since it could modify iron absorption.

4) How the adherence was evaluated?

5) 23.9% (11/46) patients abandoned the study prematurely. This number of patients leaving the study is very high, since only 5 were related to adverse events. These 5 patients were included in the safety analysis, but it is unclear what was done with the other 6 patients... Please explain in the Results section.

6) Discussion section is actually too short. There are several studies evaluating the tolerability and the efficacy of SI in different chronic diseases, and at least a paragraph about it should be added. Moreover, a recent study evaluated the effectiveness of SI in comparison with intravenous iron in a cohort of patients with UC (Bertani et al, Nutrients, 2021), and it should be commented. 

7) The inclusion of patients with different diseases (CD and UC) as well as with different disease severity (although the vast majority of patients was in remission or with mild disease) in the study population is an important limitation of the study and should be inserted in the Discussion section. Another important limitation is the high rate of patients who discontinued the study, not only related to side effects.

Round 2

Reviewer 2 Report

The authors significantly improved the paper.

However, they asked for more time in order to provide data regarding disease extension. I think that to know disease extension is very important in order to better characterize the study cohort.

Therefore, I agree in giving as much time as possible in order to provide these information.
